# Anesthetic Considerations for Patients with Hereditary Neuropathy with Liability to Pressure Palsies: A Narrative Review

**DOI:** 10.3390/healthcare12080858

**Published:** 2024-04-19

**Authors:** Krzysztof Laudanski, Omar Elmadhoun, Amal Mathew, Yul Kahn-Pascual, Mitchell J. Kerfeld, James Chen, Daniella C. Sisniega, Francisco Gomez

**Affiliations:** 1Department of Anesthesiology and Perioperative Care, Mayo Clinic, Rochester, MN 55902, USA; laudanski.krzysztof@mayo.edu (K.L.); elmadhoun.omar@mayo.edu (O.E.); kerfeld.mitchell@mayo.edu (M.J.K.); chen.james@mayo.edu (J.C.); 2School of Biomedical Engineering, Science and Health Systems, Drexel University, Philadelphia, PA 19104, USA; agm76@drexel.edu; 3St George’s University Hospitals NHS Foundation Trust, London SW17 0QT, UK; kahn-pascual@doctors.org.uk; 4Department of Neurology, University of Pennsylvania, Philadelphia, PA 19104, USA; daniella.sisniega@pennmedicine.upenn.edu; 5Department of Neurology, University of Missouri, Columbia, MO 65211, USA

**Keywords:** hereditary neuropathy with liability to pressure palsies, anesthesia, peripheral neuropathy, recommendations, standards of care, PMP22, medicolegal issues

## Abstract

Hereditary neuropathy with liability to pressure palsies (HNPP) is an autosomal dominant demyelinating neuropathy characterized by an increased susceptibility to peripheral nerve injury from trauma, compression, or shear forces. Patients with this condition are unique, necessitating distinct considerations for anesthesia and surgical teams. This review describes the etiology, prevalence, clinical presentation, and management of HNPP and presents contemporary evidence and recommendations for optimal care for HNPP patients in the perioperative period. While the incidence of HNPP is reported at 7–16:100,000, this figure may be an underestimation due to underdiagnosis, further complicating medicolegal issues. With the subtle nature of symptoms associated with HNPP, patients with this condition may remain unrecognized during the perioperative period, posing significant risks. Several aspects of caring for this population, including anesthetic choices, intraoperative positioning, and monitoring strategy, may deviate from standard practices. As such, a tailored approach to caring for this unique population, coupled with meticulous preoperative planning, is crucial and requires a multidisciplinary approach.

## 1. Introduction

Hereditary neuropathy with liability to pressure palsies (HNPP) is a heritable neuropathy characterized by recurrent episodes of focal motor, sensory, or mixed neuropathy [1]. The most common presentation of this disease process involves focal weakness and/or paresthesia in one or more peripheral nerve distributions. Episodes are often triggered by minor trauma, injury, or compression. The duration of symptoms can range from hours to months, depending on the extent of nerve injury incurred. Considering the vagueness or variability of HNPP’s clinical presentation, the illness tends to be underdiagnosed [2,3]. Recurrence is also common [4].

In this review, we aim to present the current state of understanding for this illness, with emphasis on key perioperative considerations. More than 487 manuscripts with the “HNPP; Hereditary Neuropathy with Liability to Pressure Palsies” keyword can be found on PubMed; however, there is significant repetition of manuscript contents, while the frequency of publications varies significantly on a year-to-year basis (Figure 1). This review presents current advances in genetic pathogenesis and testing as they have advanced over the past five years using the most relevant and demonstrative work. Our second goal was to demonstrate a lack of clear guidance on addressing anesthesia-related problems in patients with HNPP. Our concern is that recommendations addressing HNPP-related phenomena and care are lacking in several guidelines published by established authorities and regulatory agencies that take into consideration the general population [5]. To our knowledge, this issue has not been addressed in prior reviews. Subsequently, we review the accessible publications, focused on the most recent one in case of repetitive ones or unique in the context during preparation of this manuscript. Consequently, this manuscript is intended as a guided review with aforementioned goals in mind, not the meta-analysis. 

## 2. Etiology

HNPP is an autosomal dominant condition with a penetrance reported to vary between 70–100%. The condition also demonstrates significant variability in its observable characteristics, which makes it challenging to precisely estimate the incidence of the illness. Notably, even among monozygotic twins, variations in both observable traits and electrodiagnostic features have been observed [6]. Mechanisms underlying the emergence of HNPP secondary to mutations in PMP22 are complex and may be unique to each specific human genome [7,8]. These issues are fundamental, as they could help determine the prevalence of HNPP and potential differences in presentation and symptom severity between males and females [9]. It remains unclear if mosaicism can occur in HNPP, potentially conferring a protective effect from developing clinical symptoms while still posing a risk of clinical manifestations [10,11,12,13].

The underlying etiology for HNPP has been described as caused by abnormalities related to peripheral myelin protein 22 encoded by the namesake PMP22 gene located on chromosome 17p11.2. Approximately 20% of cases result from de novo deletions or point mutations [7,8,14,15]. HNPP can be further organized into three distinct groups. The first form results from a duplication of the PMP22 gene giving rise to Charcot-Marie-Tooth disease type 1A (CMT1A) [6,14]. A subset of said CMT1A patients exhibit a mutation: a LITAF I92V sequence that leads to accumulation of PMP22 in mitochondria, leading to cellular accumulation rather than directing it to lysosomes for proper processing [16]. 

A second form of HNPP results from a 1.5 Mb deletion in the PMP22 gene [9]. The third form is caused by point mutations in the PMP22 gene, resulting in varying clinical manifestations. These manifestations may present phenotypic features from either of the first two groups [17]. Recently, another mutation in the NEFL gene linked to the HNPP phenotype in Charcot-Marie-Tooth type 2E was described [18].

Alterations in segmental duplication sequences localized in certain parts of the genome have extensively been described as the etiopathogenesis for HNPP [19,20]. Segmental duplication sequences are described as genetic sequences sharing a high level of source identity [20]. Said sequences tend to be more than 1 kb in length, wherein several copies can be found in distinct parts of the genome. Consequently, they usually create high-complexity hubs; however their distinctiveness creates significant difficulties in fully elucidating their role in health and illnesses [21,22,23,24]. The Y and 22 chromosomes have the highest abundance of said segmental duplications. Consequently, illnesses located to the said two chromosomes more commonly emerge, as compared to other genetic illnesses [20,25,26]. Only recently has a broader view of these mechanisms been investigated in-depth and revealed in more detail, wherein they have been linked to unequal meiotic crossover and rearrangement [22]. 

Mechanistically, the pathophysiology of nerve injury in HNPP has been localized to downstream effects of decreased peripheral myelin protein 22 expression or function, quite similarly to the mechanism of CMT1A [1,27,28]. It has been posited that PMP22 products typically facilitate the production of stable and compact myelin sheaths around peripheral nerves. When aberrant, as in HNPP, irregular regions of focal hypermyelination sheets or “tomaculae” (swellings of the myelin sheath) can form [17,29]. 

Repetitive trauma (even if minor) [30] or prolonged pressure injury to the nerves results in the aggregation of said aberrant myelin, with a characteristic “onion-bulb” appearance. When subjected to compressive forces, these irregularly myelinated peripheral nerves are predisposed to exhibiting conduction blockades [29,31]. Heterozygous murine models have shown this susceptibility could be due to focal reduction in axonal diameter, secondary to the constriction by the aforementioned tomaculae, or increased myelin permeability [32,33]. Finally, Schwann cell apoptosis plays a role that needs to be better understood in the pathophysiology of HNPP nerve injury [34]. A pressure-related etiology underlying HNPP is somewhat bolstered by cases describing the emergence of HNPP after weight loss, causing loss of natural adipose nerve padding [30,35,36].

While the aforementioned mechanisms are not exhaustive, and remain to be fully elucidated themselves, they offer some insight into the broad spectrum of healing time and symptomatic improvement in these patients [4,10,12,27]. Variables at play may include the degree of underlying asymptomatic structural changes in the patient’s myelination in any particular nerve before further injury, as well as the degree and duration of compression. Lastly, many factors are involved in nerve healing or regeneration for individual patients. The specific circumstances associated with susceptibility to a prolonged injury or delayed recovery in certain patients with HNPP remain unclear. It is also unclear if and, if so, how the condition interferes with the recovery of an injured nerve. Further complicating the issue is that the patient may experience repetitive and overlapping injuries. All these three factors affect liability and responsibility issues if a debilitating injury occurs. 

## 3. Prevalence

The current reported prevalence of HNPP is in the range of 7–16:100,000; however, this may be a conservative estimate due to underdiagnosis [2,7,10,15,17,37,38,39,40,41,42,43]. Misdiagnosis or underdiagnosis may be partly attributed to the subtle clinical features and variable presentation of the disease [4,27,35,44,45,46,47,48]. Additionally, there is also international variability in reporting of HNPP, with the National Institute of Health categorizing the disease as rare, while statements from the Foundation for Peripheral Neuropathy consider it a common palsy [49,50]. In the United States, the estimated total prevalence is 200,000 individuals [3,37]. While it was initially expected that CMT1A and HNPP, both stemming from a reciprocal recombination event on chromosome 22, would exhibit similar disease prevalence [28], recent studies suggest otherwise. CMT1A is now identified as the most prevalent heritable neuropathy, with an incidence of 10.61 per 100,000 (95% CI 7.06–14.64), constituting a substantial portion of the pooled incidence of 17.69 per 100,000 (95% CI 12.32–24.33) (*p* < 0.001) [51].

## 4. Diagnosis and Presentation

### 4.1. Clinical Manifestations 

HNPP typically presents in the second to third decade of life with variable symptoms that may be protean in presentation [1,4,10,12,13,17,27,35,37,46,52]. The disease is characterized by acute recurrent neuropathies of sensory, motor, or mixed variants in one or more peripheral nerve distributions [13,17,27,35,44,46,47,53,54]. The most common presentation involves focal, non-painful, and motor mononeuropathies, resulting in weakness or paresthesia [27]. Most episodes are provoked by compression, traction, or minor trauma and are followed by complete recovery within days to months. However, a minority of patients may have persistent disability, which is usually mild. Persistent neurologic deficits, such as hand weakness or foot drop, are uncommon but possible [55]. Additionally, some individuals with HNPP may exhibit mild-to-moderate peripheral neuropathy or autonomic dysfunction. 

Symptomatic episodes are usually characterized by weakness, sensory changes, reduced deep tendon reflexes, muscle cramps, fatigue, frequent falls and/or leg and ankle swelling [56]. Painful manifestations are variable but less common features [44]. In some cases, pain may lead to decreased physical activity [45]. Musculoskeletal and neuropathic pain can potentially be primary symptoms at the time of initial presentation [27]. The link between HNPP and pain can be explored through four potential different mechanisms. First, given its neuropathic etiology, HNPP may involve neuropathic pain mechanisms. Second, altered central processing (or central sensitization) may be linked to the underlying neuropathic process. Third, local tissue stress from prolonged weakness or secondary joint deformity can contribute to peripheral nociception, resulting in musculoskeletal pain. Lastly, it is possible that co-occurrence of HNPP and pain could be coincidental. In addition, myoclonus and restless leg syndrome have also been noted to be presenting factors [48].

Episodes can be spontaneous or precipitated by compression, shear forces, and minor local trauma. Certain prolonged positions, such as flexion and extension of limbs, are potential triggers, highlighting the importance of careful intra-operative positioning for these patients, especially during prolonged surgical procedures [17,35,46,53]. HNPP primarily affects nerves that are vulnerable to compression due to their anatomical location relative to bony prominences, and particularly the radial, median, ulnar, and common fibular nerve fibers [53,57,58]. While less frequently affected, the facial nerve has been previously described as also liable to pressure palsy [58]. As such, common sites include, in descending order of frequency: peroneal nerve entrapment at the fibular head leading to foot drop, ulnar nerve compression at the elbow with hypothenar and interossei paresis, classical carpal tunnel syndrome, and brachial plexus injuries [54,57,58,59,60,61]. Chronic or prolonged entrapment may also cause muscle wasting, such as seen in carpal tunnel syndrome [30,46,59]. Depending on the degree of nerve injury, the duration of episodes varies from hours to months [13,62]. Nearly 50% of episodes can progress to full recovery, whereas 50% may have persistent deficits associated with an episode [58]. There is a paucity of studies regarding the quality-of-life impact of HNPP, however, quality of life has been reported as similarly impaired in both HNPP and CMT1A [44,45,48,56,63]. On physical examination, deep tendon reflexes may be altered (most commonly diminished) [27]. However, as manifestations are varied and dependent on the location of the affected nerves, not all signs and symptoms may coincide together [27]. Rarely, cognitive decline and diminished grey and white matter volume can occur [64]. HNPP should be considered as a potential diagnosis in individuals experiencing recurrent episodes of transient motor or sensory neuropathy, particularly in anatomically vulnerable areas, where common causes seem unlikely. A strongly suggestive family history of similar recurrent presentations is also strongly supportive, given the autosomal dominant inheritance [47]. However, it is not a foregone conclusion to diagnose HNPP in all cases when one family member carries the diagnosis, due to the variable penetrance of gene mutation [3,16,17,26,27,65]. The differential diagnosis for HNPP includes compression neuropathies, diabetic polyneuropathy, vasculitis polyneuropathies or multifocal mononeuropathies, and chronic inflammatory demyelinating polyneuropathy or hereditary neuralgic amyotrophy [50,52,66]. Of note, there can be an overlap between the above pathologies, which should be considered as a confounding factor. 

### 4.2. Diagnosis

#### 4.2.1. Electrophysiological and Imaging Findings

Electrophysiological studies may demonstrate a prolongation in motor and sensory nerve conduction latencies measured at the median and common fibular nerves distal to compression sites (wrist and fibular head, respectively). Both symptomatic and asymptomatic individuals may display this pattern on said evaluations [27,59,61,67,68,69,70]. Bilateral median sensorimotor latencies, and at least unilateral peroneal motor conduction abnormality, is considered very suggestive for the disease, and one-third of these patients may exhibit diminished or abolished sural responses [71].

Ultrasound and MRI evaluation may also contribute to suggestive findings [29,30,61,72]. Indeed, nerve sonography has been utilized in the past, describing a difference between CMT1A, wherein nerve cross-section was noted as increased, as opposed to an axonal CMT2A, and wherein fascicle diameter was increased, leading the authors to propose nerve ultrasonography as a tool for differentiating between heritable neuropathies [68]. Advanced MRI techniques have shown promise in the diagnosis of HNPP and CMT. Diffusion tensor imaging has shown white matter changes in HNPP patients with a tendency to affect frontal lobes [73]. One small case series obtained high MRIs of HNPP and CMT1A patients when evaluating for magnetic transfer ratio (an indirect measure of myelination) and nerve cross section; morphological circularity of the nerves proposed said measures as reliable and sensitive for CMT1A only [74]. Fatty infiltration of muscle in a proximal dependent fashion has been both suggested as further evidence of a length dependent process and as a biomarker of HNPP disease progression in general [71].

#### 4.2.2. Genetic Testing

The confirmation of HNPP diagnosis occurs in an individual displaying indicative clinical and electrophysiological observations, along with the detection of either the 1.5-Mb recurrent deletion or a new deletion affecting PMP22 (in 80% of cases), or a PMP22 sequence variation (in 20% of cases) identified through molecular genetic analysis. This is usually accomplished using chromosomal microarray, fluorescent in situ hybridization (FISH), or polymerase chain reaction (PCR) amplification, revealing either a 1.5 Mb deletion or point mutations involving the PMP22 gene [9,14,18,26,38]. As more advanced molecular genetic diagnostics become available, there is potential for earlier detection. However, clear indications for pre-emptive screening still need to be defined, as testing remains to be widespread or economical, and the relative societal burden is small. Genetic testing is often deployed as a confirmatory tool for patients with strongly suggestive symptoms.

## 5. Treatment

Treatment strategies for HNPP primarily involve addressing symptoms through occupational therapy and physical therapy tailored to improve both fine motor and gross motor skills, essential for daily activities. Bracing, such as wrist splints or ankle-foot orthoses, might offer temporary or permanent assistance. Specialized footwear, particularly those providing adequate ankle support, may prove necessary. Neuropathic pain management often entails analgesic medications, such as acetaminophen and NSAIDs or gabapentin. Additionally, protective padding at pressure points like elbows or knees can mitigate nerve damage from pressure or trauma. Evidence for surgical treatment remains limited to case reports, with no large-scale studies available [75]. Specifically, the effectiveness of carpal tunnel release surgery in patients with HNPP and carpal tunnel syndrome remains unclear and necessitates careful patient selection. Case reports of surgical success have included bilateral carpal and cubital tunnel releases; ulnar entrapment release and tendon transfer procedures may also effectively improve patient functional status [61,75].

The illness may be very debilitating, even if it is commonly underrecognized or recognized late in life [1,13,44,47,63]. Thus, regular neurological assessments, emphasizing muscle condition, strength, sensory perception, and neuropathic discomfort, are crucial. Physical and occupational therapy evaluations focus on refining motor skills and aiding daily tasks, while foot examinations guard against pressure sores and ill-fitting footwear [76]. Hence, it becomes clear that the focus on perioperative risk minimization is paramount. 

## 6. Anesthetic Management

Currently, there are no established, widely adopted guidelines for managing HNPP patients in the perioperative period. Regardless, general recommendations can be made based on the available data. Considering complexity, heterogeneity, and variable presentation of the HNPP in peri-operative period, an in-depth discussion with patients is warranted to delineate risks and to describe mitigating procedure, or whether the risk-to-benefit ratio is worth pursuing the procedure at all. Overall, the risk of HNPP-related perioperative neuropathy is difficult to ascertain as we do not know the denominator. This denominator may prove quite elusive, considering that clinical presentations vary. 

During the pre-anesthesia evaluation, clinicians should focus on risk mitigation and identify the patient’s specific triggers and develop strategies to minimize HNPP episode incidence or recurrence. Patient-related risk factors for perioperative neuropathy (as extrapolated from the general population) should be noted including, but not limited to, alcohol and tobacco use, obesity or being underweight, status, vascular disease, and advancing age. Furthermore, operation-related risk factors include positioning and procedure length [77]. Notably, several authors have noted recent weight loss as an inciting factor for symptom development [78]. Multidisciplinary discussions can aid in treatment planning when anticipated deviation from the standard of care is necessary. Patient involvement in this process, and implementing the “shared consent” concept, seems prudent [57,79]. Documenting any pre-existing neuropathies also seems prudent practice. A summary of NHPP-specific considerations can be found in Appendix A. 

### 6.1. Pre-Induction Period

Preventative strategies should focus on risk mitigation, as mentioned above, including protecting areas most vulnerable to compression nerve injury, such as the wrists, elbows, knees, hips, and ankles. Injury from routinely utilized anesthetic equipment, such as noninvasive blood pressure cuffs, should be foreseen and minimized. Care should be taken to ensure ample padding and to avoid positioning extremities in excessive flexion or extension for protracted periods of time, as well as routine passive range of motion, when able to, on exposed extremities [80]. In prone positioning cases, limbs should be wrapped in quilted cloth and protected with gel pads, paying attention to the nerves most susceptible to damage. For procedures lasting longer than three hours, incorporating ten-to-fifteen-minute breaks every hour to reassess and readjust limb position can be highly beneficial, as described below [79].

### 6.2. Maintenance of Anesthesia

Limited evidence exists regarding the optimal anesthetic choice for HNPP patients. A singular case report suggests that peripheral or neuraxial regional blockade techniques are safe and unlikely to worsen the patient’s peripheral neurological function [81]. However, using the lowest concentration and volume of local anesthetic necessary is advised to minimize potential risks in this vulnerable patient cohort [82]. Due to limited data, avoiding peripheral nerve blocks is recommended unless necessary for safe anesthetic delivery. This recommendation is based on the mitigation of medicolegal risk in a situation where the source of the nerve palsy cannot be ascertained between operators involved in patient care, due to the nature of surgery (carpal tunnel syndrome, spine surgery, other procedures with an elevated risk of surgical nerve injury) [68].

Thorough neurological assessment before and after each anesthetic is crucial to identify preexisting deficits and detect new symptoms. This documentation can also benefit future anesthesia or surgical providers when planning procedures or anesthetic approaches [57]. 

Some reports suggest increased sensitivity to specific neuromuscular blocking agents, namely rocuronium and succinylcholine, with potentially prolonged blockade [83]. Therefore, carefully monitoring neuromuscular blockers, minimizing exposure, and using the appropriate dose are crucial to minimize this risk. Of note, the molecular basis of altered sensitivity to neuromuscular blocking agent is difficult to ascertain and may vary between different presentations of this illness.

### 6.3. Temperature 

Extremes of temperature have been shown to worsen neuropathies due to structural differences in nerve anatomy in these patients, specifically myelin, which usually serves a role in effective conduction and axon protection. While specific temperature cutoffs remain unidentified, maintaining normothermia in the perioperative period is advisable [84]. Additionally, advising patients to use gloves and wear protective clothing in cold environments may help minimize the risk of exacerbations of HNPP symptoms due to cold exposure post-operatively [76,85]. Combined with preoperative counseling, this measure could be helpful in mitigating nerve injury. 

### 6.4. Equipment and Monitoring

Standard blood pressure cuff inflation can cause significant discomfort in these patient cohorts during monitored anesthesia care (MAC) and can potentially lead to permanent nerve damage. Early blood pressure cycling frequency modification in response to patient discomfort can help mitigate postoperative neuropathy. However, this may not be possible in deeply sedated patients. The strategy for intraoperative blood pressure monitoring in these complex patients requires a tailored approach to each patient. Providers may consider using noninvasive cuffs with less frequent cycling or opt for invasive intra-arterial monitoring. That said, the risks and benefits of each approach should be weighed for each patient. For example, direct or indirect nerve injury during placement of invasive arterial lines should not be ignored. To this extent, ultrasound use for arterial line placement should be mandatory in this cohort. If the planned procedure warrants arterial line placement, the concurrent use of a noninvasive blood pressure cuff, as is often done for cross-referencing, could conceivably be curtailed after an initial validation check.

Patients should also be informed about the increased risk of common fibular nerve injury from using sequential pneumatic compression devices (SCDs) for deep venous thromboembolism prevention. Careful adjustments to the position of said SCDs and avoiding the fibular head on application may reduce the risk of injury. Although cranial nerve injury is rare in HNPP, case reports have described trigeminal, facial, and hypoglossal nerve injury [66]. A sole case report involved phrenic and laryngeal injury in a confirmed HNPP patient, which the authors described as exceptionally rare [78]. The more superficial branches of the zygomatic, buccal, and mandibular nerves are at risk of compression injury from bag-valve-mask ventilation or excessively tight endotracheal tube (ETT). Care should be taken to minimize this risk, with padding, taping versus tying ETT, or frequent repositioning of ETT fasteners. 

Providers may consider the deployment of intraoperative electrodiagnostic monitoring techniques, such as motor and somatosensory evoked potentials (SSEP), in expectedly protracted cases [29,67,68,69,70]. SSEP can be useful for early detection of emerging peripheral nerve injury. Prior assays in spine and skull surgery found a higher incidence of SSEP abnormalities attributable to prone or lateral decubitus positioning. More importantly, intraoperative adjustments to patient position in response to real time SSEP data have been found to prevent or mitigate postoperative peripheral nerve injury [68,69,70]. SSEP may be applicable for intraoperative monitoring in HNPP patients with a higher risk of perioperative neuropathy [77]. However, at the time of writing, there is limited evidence to support said technique in this specific condition. 

It is important to carefully assess and document baseline conduction deficits that are present preoperatively. This would allow for differentiation between preexisting and de novo injuries during surgeries. Close collaboration with neurophysiology teams would facilitate the safe implementation of this monitoring tool [57]. 

### 6.5. Regional Anesthesia 

There is a paucity of studies, or even case reports, examining the use of regional anesthesia in HNPP. One such case report involved epidural anesthesia via low-concentration bupivacaine and fentanyl administered in labor to no ill effect [81]. Another report involved two deliveries in the same patient, who received epidural bupivacaine and sulfentanil [86]. Similarly, other authors administered epidural bupivacaine for arthroscopic anterior cruciate ligament reconstruction successfully [87]. As mentioned above, there are no randomized control trials for general vs. regional anesthesia in HNPP, however, there may be some benefits derived from decreased operating and recovery room time [55].

### 6.6. Positioning 

Intraoperative positioning has been described as a preventive or mitigating measure for neuropathy in the general population [80,87]. Thus, in HNPP patients who are particularly vulnerable to compression or shearing injury, it would be logical to apply measures in order to mitigate said risk exposure, although they need to be extrapolated from recommendations concerning the general population, as the evidence-based guidance is lacking. In case of upper extremities: avoidance of prolonged elbow flexion more than 90 degrees, limiting arm abduction in supine patients to less than 90° [80]. In supine patients, neutral or supinated forearm positioning may be preferable and avoidance of pressure upon the post-condylar humeral groove is recommended if using an arm board [87]. As for the lower extremities, one should limit pronounced hip flexion or extension. Avoid wherever possible extended pressure on the peroneal head [80,87]. Furthermore, noting patient positioning during surgical stages may be useful for identifying precipitating events. 

### 6.7. Medicolegal Aspects

Despite careful and proactive attempts to mitigate any postoperative morbidity in this challenging patient cohort, complications may still occur. In the US, peripheral nerve injury is reported as the third most common cause of litigation regarding anesthesia malpractice, although systematic review has reported said claims as waning [87].

Engaging patients in shared decision-making, and thorough risk benefits regarding the anesthesia plan, may encourage greater patient involvement and potentially mitigate the risk of legal action against providers. The shared decision-making discussions and the final anesthesia plan chosen should be carefully documented in the patient’s chart. The process allows the medical team to establish rapport with the patient, a process itself that lowers the risk of future litigation.

Considering necessary elements for litigation, deviating from the standard of care is the basis of most arguments presented by the plaintiff. The Standards for Basic Anesthetic Monitoring, as approved and reaffirmed by the American Society of Anesthesiologists, state: “Every patient receiving anesthesia shall have arterial blood pressure and heart rate determined and evaluated at least every five minutes. *” At first glance, this appears to restrict a modified anesthetic plan, but in this published standard, the ASA further codified that “Under extenuating circumstances, the responsible anesthesiologist may waive the requirements marked with an asterisk (*)”; it is recommended that when this is done, it should be stated in the patient’s medical record, to clarify any rationale behind deviation from standard practice [5,88].

## 7. Conclusions

Patients with hereditary neuropathy with liability to pressure palsies (HNPP) present a unique and challenging population for the perioperative team. Their vulnerability to incapacitating nerve damage and other complications necessitates careful multidisciplinary and meticulous preoperative planning. 

While awaiting robust large-scale studies on best practices for caring for HNPP patients, the measures described in this paper, emphasizing pre-operative identification and planning of patient positioning strategies to mitigate risk or damage, may be crucial to improving their perioperative outcomes. 

## Figures and Tables

**Figure 1 healthcare-12-00858-f001:**
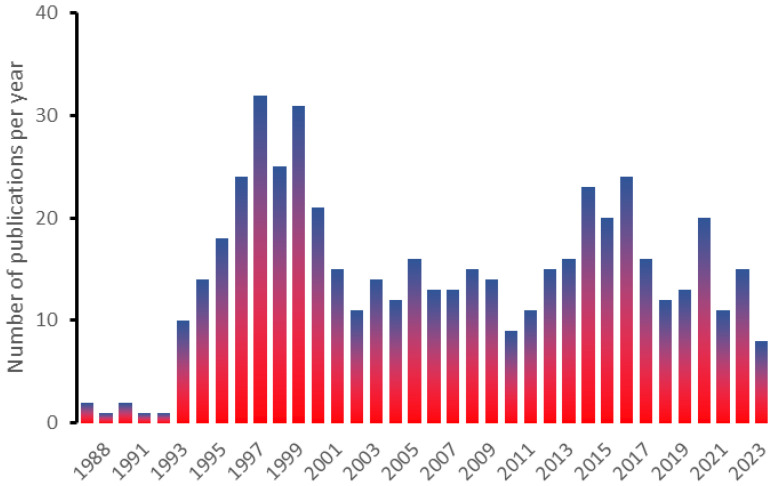
This histogram shows the frequency of manuscripts found with the HNPP keyword on PubMed (1988–2023).

## Data Availability

Not applicable.

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
