# Peer review of "Anesthetic Considerations for Patients with Hereditary Neuropathy with Liability to Pressure Palsies: A Narrative Review"

_healthcare, 2024, doi:10.3390/healthcare12080858_

Round 1
Reviewer 1 Report
Comments and Suggestions for Authors
This review summarized the scientific evidence about HNPP
The research presented in the article certainly belongs to the Journal topics. I thank the authors for their effort; this is an important, original and clinically interesting topic
The Abstract is very concise which is good, but lacks at least the most important statistical data such as the number and type and the number of articles analyzed.
Please describe the “Search string” used and the scientific literature databases used: L.47 “More than 487 manuscripts with HNPP keywords can be found on PubMed” , Why were only the mentioned databases used by the authors in their search? A more detailed description is needed regarding the included studies
The title should be changed into narrative or scoping review
The main aim of this review is: “The purpose of this review is to present contemporary evidence and recommendations for optimal care for HNPP patients in the peri- operative period.” , this is the main question in the Abstract, but in the text more section are reported , they should be supported by relevant literature and should be presented in the abstract
L.147 “Symptomatic episodes are usually characterized by muscle cramps, fatigue, frequent falls and/or leg and ankle swelling [54].” Clinical examination in HNPP may reveal muscular weakness, atrophy, sensory signs, reduced deep tendon reflexes: please add recent literature evidence about the correlation of some postural and gait parameters with muscular weakness, reduced deep tendon reflexes , risk of fall. Postural and gait impairments may play an important role in HNPP complications, but dedicated screening methods and treatment strategies are still lacking (Include more recent references, highlighting the association between postural alterations and neuropathy, such as DOI: 10.1016/j.jtv.2023.10.002)
Comments on the Quality of English Languagemoderate editing required
Reviewer 2 Report
Comments and Suggestions for Authors
"Authors reviewed the anesthesia options and perioperative considerations in patients with HNPP, a rare neurological disorder.
Title is relevant to the text. Abstraction of the text is well enough. Keywords are relevant. The review is scientific and likely to fullfill a gap in the medical literature. However, several revisions may improve the manuscript. - Recommendations for Clinical Manifestations section: Authors should consider providing additional data. Hereditary neuropathy with liability to pressure palsies (HNPP) is marked by recurring episodes of acute sensory and motor neuropathy affecting one or more nerves. Typically, it begins with the sudden onset of painless focal sensory and motor neuropathy in a single nerve (mononeuropathy). While the first occurrence typically arises in the second or third decade of life, earlier onset is feasible. Neuropathic pain is increasingly acknowledged as a prevalent symptom. Generally, complete recovery follows acute neuropathy; however, in cases where recovery is incomplete, resultant disability tends to be mild. Additionally, some individuals with HNPP may exhibit mild-to-moderate peripheral neuropathy. - Recommendations for Diagnosis section: Diagnosis of the disease can be enriched by stating the following: The confirmation of HNPP diagnosis occurs in an individual displaying indicative clinical and electrophysiological observations, along with the detection of either the 1.5-Mb recurrent deletion or a new deletion affecting PMP22 (in 80% of cases), or a PMP22 sequence variation (in 20% of cases) identified through molecular genetic analysis. - Recommendations for Treatment section: Authors somewhat stated the management options of the disease. However, additional data could be provided as follows: Treatment primarily involves addressing symptoms through occupational therapy and physical therapy tailored to improve both fine motor and gross motor skills, essential for daily activities. Bracing, such as wrist splints or ankle-foot orthoses, might offer temporary or permanent assistance. Specialized footwear, particularly those providing adequate ankle support, may prove necessary. Neuropathic pain management often entails analgesic medications. Additionally, protective padding at pressure points like elbows or knees can mitigate nerve damage from pressure or trauma.Regular neurological assessments, emphasizing muscle condition, strength, sensory perception, and neuropathic discomfort, are crucial. Physical and occupational therapy evaluations focus on refining motor skills and aiding daily tasks, while foot examinations guard against pressure sores and ill-fitting footwear.
Certain activities and conditions, such as prolonged leg crossing or leaning on elbows, repetitive wrist movements in certain occupations, rapid weight loss, or vincristine use, should be approached with caution due to their potential to exacerbate symptoms.
Asymptomatic relatives with a familial risk may consider genetic testing for the specific PMP22 variant identified in affected family members to determine their status and receive guidance on avoidance measures and lifestyle adjustments. - Conclusions are justified, and I am agree with the authors about the conclusions drawn. I summary, the review paper is well enough to be published in the journal, however, several revisions must be considered as stated above.".Author Response
please see attached file

Round 2
Reviewer 1 Report
Comments and Suggestions for Authors
Author response missing in a crucial points , remain several configurations that do not fit the format of medical journals.
such as
"Please describe the “Search string” used and the scientific literature databases used: L.47 “More than 487 manuscripts with HNPP keywords can be found on PubMed” , Why were only the mentioned databases used by the authors in their search?
A more detailed description is needed regarding the included studies"
other suggestions useful for improving the paper were also not considered:
l.153-154 references suggested in order to better describe the association between postural alterations and neuropathy (DOI: 10.1016/j.jtv.2023.10.002) was not considered by authors
Comments on the Quality of English Languageminor editing
Author Response
We are somewhat puzzled by receiving a reply a couple of hours after submitting the manuscript. However, we appreciate the fast turnaround.
"Please describe the “Search string” used and the scientific literature databases used: L.47 “More than 487 manuscripts with HNPP keywords can be found on PubMed.” Why did the authors use only the mentioned databases in their search?]
- This manuscript is a review, not an analysis of all possible data. The string is embedded in the body of the manuscript. We added sentences signifying why we picked a certain publication not all of them.
“l.153-154 references suggested in order to better describe the association between postural alterations and neuropathy (DOI: 10.1016/j.jtv.2023.10.002) was not considered by authors” AND “other suggestions useful for improving the paper were also not considered” AND “A more detailed description is needed regarding the included studies”
- We addressed all the reviewers' suggestions. A detailed table describes our responses to the remarks. In the process championed by several co-authors, we decided to implement the reviewers’ remarks that we found most contributing. However, we would like to reserve the right to control the manuscript content with the guidance of the reviewers.
- The reference suggested by this reviewer title is “Use of honey in diabetic foot ulcer: Systematic review and meta-analysis” by Karadeniz and Serin. I reviewed this study, but I cannot find the connection to HNPP. We added mentioning about autonomic dysfunction to text.
- Three authors have described the studies to reflect the contents and essence of the cited article. This reviewer's suggestion is vague and lacks specificity. It is also somewhat contradictory to his own evaluation of the manuscript in this reviewer's 3/5 stars or even higher eval by another reviewer.